# Effects of Consuming Heat-Treated Dodamssal Brown Rice Containing Resistant Starch on Glucose Metabolism in Humans

**DOI:** 10.3390/nu15102248

**Published:** 2023-05-09

**Authors:** Jiyoung Park, Sea-Kwan Oh, Miae Doo, Hyun-Jung Chung, Hyun-Jin Park, Hyejin Chun

**Affiliations:** 1Department of Central Area Crop Science, National Institute of Crop Science (NICS), Rural Development Administration (RDA), 126 Suin-ro, Kwonseon-gu, Suwon 16429, Republic of Korea; pjy2812@korea.kr; 2National Institute of Crop Science (NICS), Rural Development Administration (RDA), Wanju 55365, Republic of Korea; seakwanoh@gmail.com; 3Department of Food and Nutrition, Kunsan National University, Gunsan 54150, Republic of Korea; miae_doo@kunsan.ac.kr; 4Division of Food and Nutrition, Chonnam National University, Gwangju 61186, Republic of Korea; hchung@jnu.ac.kr; 5Department of Biotechnology, College of Life Sciences and Biotechnology, Korea University, 145 Anam-ro, Seongbuk-gu, Seoul 02841, Republic of Korea; hjpark53@korea.ac.kr; 6Department of Family Medicine, Ewha Womans University College of Medicine, Seoul 07804, Republic of Korea

**Keywords:** Dodamssal, functional food, glucose metabolism, glycemic control, metabolic syndrome, obesity, *Oryza sativa*, resistant starch, high amylose rice, rice cultivar

## Abstract

Rice is a major source of carbohydrates. Resistant starch (RS) is digested in the human small intestine but fermented in the large intestine. This study investigated the effect of consuming heat-treated and powdered brown rice cultivars ‘Dodamssal’ (HBD) and ‘Ilmi’ (HBI), with relatively high and less than 1% RS content, respectively, on the regulation of glucose metabolism in humans. Clinical trial meals were prepared by adding ~80% HBI or HBD powder to HBI and HBD meals, respectively. There was no statistical difference for protein, dietary fiber, and carbohydrate content, but the median particle diameter was significantly lower in HBI meals than in HBD meals. The RS content of HBD meals was 11.4 ± 0.1%, and the HBD meals also exhibited a low expected glycemic index. In a human clinical trial enrolling 36 obese participants, the homeostasis model assessment for insulin resistance decreased by 0.05 ± 0.14% and 1.5 ± 1.40% after 2 weeks (*p* = 0.021) in participants in the HBI and HBD groups, respectively. The advanced glycation end-product increased by 0.14 ± 0.18% in the HBI group and decreased by 0.06 ± 0.14% in the HBD group (*p* = 0.003). In conclusion, RS supplementation for 2 weeks appears to have a beneficial effect on glycemic control in obese participants.

## 1. Introduction

Obesity is an epidemic that causes metabolic disorders including insulin resistance and diabetes [1]. In addition to providing the main nutrients for survival, functional foods have the advantage of preventing chronic diseases, such as the abovementioned. Consumer demand for high-quality products containing health functionalities has influenced the development of new nutrients and modern food processing technologies [2].

Rice is a common source of carbohydrates in the human diet as the rice grain is composed primarily of starch (75–80%), approximately 12% water, and only 7% protein [3]. Starch is classified into rapidly digested starch (RDS), slowly digested starch (SDS), and undigested or resistant starch (RS) based on its digestibility [4]. Among them, RS is not digested in the human small intestine, but it is fermented by microorganisms in the large intestine, thus providing various physiological benefits, such as sugar and lipid metabolism, intestinal health, and weight loss [2]. As a dietary fiber, RS is fermented by colon microorganisms in the large intestine to produce short chain fatty acids, which have the effect of improving glucose metabolism and lipid metabolism [2]. Various factors affect the formation of RS, including the unique properties of starch (starch crystallinity, granular structure, ratio of amylose to amylopectin, chain length, etc.), processing or storage conditions such as heat treatment, and interactions between starch and other components [5].

Starch digestibility is useful for predicting the glycemic response to ingested starchy foods by determining the expected glycemic index (eGI) [6]. The eGI is a ranked-measure of polysaccharide-based foods according to their ability to raise blood sugar levels when compared to a control, such as white bread or glucose. The GI is typically determined using an in vivo method and relates well to the actual food digestion process; however, the method is very complex since it involves supervision of the subject, who may respond differently to certain stimuli. In addition, the in vivo method is expensive and cumbersome to perform. Thus, a simpler in vitro method can be performed to simulate the actual food digestion process [7]. Generally, rice is regarded as a high-GI food, but this varies depending on the rice variety, grain make-up, and processing methods [7,8]. Starch particle size also affects the digestibility and GI [8,9].

Chinese researchers selected an indica rice line with increased RS content from mutational breeding using irradiation, and reported that the starch particles were round and small [10]. In Korea, a japonica rice variety named Dodamssal, which has high amylose and RS content (≥10%), was developed by conventional breeding from Goami (a breed with an amylose content of 26.7%) and Goami2 (a mutant breed developed using N-methyl-N-nitrosourea treatment), and it contained resistant starch and approximately 33% of amylose. We investigated the quality of this rice variety as a low-GI and prebiotic material with C-type starch crystallinity [11]. We also conducted studies on the effects of glucose metabolism and lipid metabolism in mice [12]. Owing to its high gelatinization temperature and low viscosity, this rice variety has a different texture from the rice commonly consumed in Korea; therefore, it is rarely used as cooked rice and is recommended for processing. Therefore, we investigated the rice powder characteristics, extruded rice cracker properties, and heat treatment processing properties of Dodamssal [13,14,15]. To enhance the functionality of the high RS-containing brown rice cultivar Dodamssal, we used a known steaming and roasting heat treatment method to maximize the RS content and lower the GI [16]. We hypothesized that the raw material could be used in powdered meals. Meals in powder form are simple and universal meal-replacement products for busy consumers, and also have advantages such as nutrition and health supplements to target consumer needs [17].

One reason for the heightened interest in RS is the remarkable increase in the prevalence of overweight and obese populations over the past 30 years, which are increasing not only in Western countries but also in Korea, where it is becoming an important public health problem [18,19]. Moreover, when using microsimulation to estimate the prevalence of obesity among adults aged 19 and older in Korea, it is expected that 70.05% (81.23% males and 59.07% females) will be categorized as “preobese” with a body mass index (BMI) of 23–24.9 kg/m^2^ by 2040, while 24.88% of all adults will be obese [20]. Obesity is a silent killer that significantly increases the risk of various diseases, such as hypertension, dyslipidemia, type 2 diabetes mellitus (T2DM), coronary heart disease, cerebrovascular disease, gallbladder disease, arthropathy, polycystic ovary syndrome, sleep apnea, and some neoplasms [21]. Along with the prevalence of obesity, the estimated prevalence of diabetes in Korean adults was 16.7% in 2020, which is very high [22]. Considering these national statistics, it is necessary to improve the continuous and comprehensive management, in addition to continuous assessments and to raise public awareness of diabetes.

Despite the increased interest in RS, little data are available on its effects on glucose metabolism in obese human populations. Therefore, this study was conducted to investigate the effect of consuming the rice cultivar Dodamssal, which contains RS, on the regulation of glucose metabolism through clinical trials in humans.

## 2. Materials and Methods

### 2.1. Sample Preparation

The rice cultivars used in the clinical trial meal were Ilmi, which has medium amylose and negligible RS (≤1%) contents, and Dodamssal, which has relatively high amylose and RS content. Based on our previous studies [16] investigating the heat treatment conditions that yield the greatest increase in RS and decrease GI in Dodamssal, we steam washed rice at 95 °C for 30 min, dried it according to the method proposed by Park et al. [16], and then roasted it at 240 °C for 10 min. The heat-treated brown rice cultivars were pulverized and used as raw materials to prepare the two clinical trial powder meals, which were manufactured by the Ricebiotec Corporation (Gyeongsan, Gyeongsangbuk-do, Republic of Korea), a rice-processing company.

We analyzed the calorific content of the two heat-treated rice samples as the main raw material for the clinical trials. Based on the calorific content of heat-treated brown rice ‘Ilmi’ (HBI; 393 kcal/100 g) and heat-treated brown rice ‘Dodamssal’ (HBD; 405 kcal/100 g), the calorific content was matched in the two meals by adding 44.5 g of HBI and 43 g of HBD, comprising ~80% of each brown rice powder. Additionally, to enhance the texture, equal amounts of soybean powder, roasted oat powder, salt, and enzyme-treated stevia were added, and the HBI meals and HBD meals were produced as powdered meal clinical trial products.

### 2.2. Composition and Particle Size Distribution Analysis

The HBI and HBD powdered samples and clinical trial meals used in this study were analyzed for nutritional composition according to the Korean Food Code [23]. Calorific content and each nutrient component are expressed in kcal and g per 100 g, respectively.

The particle size distribution of the HBI and HBD powdered samples and clinical trial meals was determined using a particle size analyzer (Mastersizer 2000; Malvern Instruments Ltd., Malvern, UK) with ethanol as the solvent.

### 2.3. RS Content and In Vitro Starch Digestibility

RS content and starch digestibility were determined according to a previous study [11]. An RS assay kit (Megazyme International Ireland Ltd., Wicklow, Ireland) and the method described by the Association of Official Agricultural Chemists were used. Sodium acetate buffer (1.2 M, pH 4.5) and pancreatin α-amylase were added to the sample (100 mg), and the mixture was incubated at 37 °C for 16 h. The precipitate was dispersed and dissolved in 2 M KOH. Subsequently, 1.2 M sodium acetate buffer (pH 3.8) and amyloglucosidase were added, and the mixture was incubated at 50 °C for 30 min. The conversion of the RS content to D-glucose was based on the amount of starch hydrolyzed. Starch digestibility was used to determine the hydrolysis index (HI) and estimated GI (eGI) of the samples. To calculate starch digestibility, the total starch content of each sample was determined using a Megazyme Total Starch Assay Kit (Megazyme International Ireland Ltd.). Porcine pancreatic-amylase (P7545; Sigma-Aldrich Corp., St. Louis, MO, USA) was dispersed in distilled water with 100 mg of the sample. The mixture was centrifuged at 1500× *g* for 10 min, and the collected supernatant was mixed with 0.3 mL of amyloglucosidase (A9913; Sigma-Aldrich Corp.). Subsequently, 4 mL of sodium acetate buffer (pH 5.2) was added to 1 mL of the enzyme mixture and 5 glass beads, and the mixture was stirred at 1500× *g*. After 180 min, the reaction product was mixed with 80% (*v*/*v*) ethanol. Finally, the glucose content was determined using glucose oxidase and peroxidase assay kits (Megazyme International Ireland Ltd.). The total starch content and ratio of the areas of the rice flour and standard material (white bread) digestibility curves were used to calculate the HI. The eGI was obtained using the equation as follows: eGI = 39.71 + 0.549 HI.

### 2.4. Clinical Trial

#### 2.4.1. Study Design and Participants

This was a double-blind, placebo-controlled clinical trial using HBI and HBD meals. A total of 36 obese participants were enrolled in this study through advertisements in the Department of Family Medicine, CHA University Bundang Medical Center, Seongnam City, Gyeonggi-do, Korea. Three participants were excluded from the study for at least one of the following reasons: (1) taking medications that may affect weight (appetite suppressants, absorption inhibitors, antidepressants, etc.), diuretics, contraceptives, steroids, hormones, or lipid-lowering agents; (2) taking medications for hyperthyroidism or hypothyroidism; (3) consuming health supplements or herbal medicines for weight control within 4 weeks of visit; (4) uncontrolled hypertension; (5) type 1 diabetes mellitus or T2DM; (6) a fasting blood sugar level of 126 mg/dL or higher in a screening test; (7) impairment of liver or renal function; (8) diagnosed with cancer; (9) those with hypersensitivity to the test food; and (10) pregnant and/or breastfeeding women. The clinical characteristics of the study participants are summarized in Table 1. Each participant signed a written consent form to participate in this study, and were randomly assigned to either the test or control group in the order of registration after going through the run-in period. Until the end of the study, the researchers and participants were not disclosed to which group they were assigned. This clinical trial was approved by the Institutional Review Board (No: 2018-11-011) of the CHA University Bundang Medical Center.

The primary goal of this study was to investigate whether Dodamssal consumption improved glucose metabolism in obese individuals. The secondary outcome was to confirm whether metabolic profiles related to obesity, including total body fat, waist circumference, visceral fat area, lipid profiles, liver enzymes, and inflammatory markers improved together.

All participants received three half-cooked meals and snacks every day for two weeks by delivery. They were required to refrain from consuming any food other than the delivered food, to which they agreed during the informed consent form process; however, water intake was not restricted. They were divided into two groups and administered HBI or HBD in powder form along with other powder-supplemented meals. As shown in Table 2, there were no differences in the carbohydrate and fiber content of the HBI and HBD powder. In summary, there was no difference in the calorific content or menu composition, other than the fact that the HBD group participants received an RS content of 19.6 g/day. Powdered HBI and HBD meals were dissolved in 180–220 mL of water and administered twice daily. The daily calorific intake of the delivered food, including HBI or HBD powder, was 1800 kcal. Moreover, since the primary outcome was to improve glucose metabolism and not weight loss, a low-calorie diet (1000–1200 kcal/day) or very low-calorie diet (<800 kcal/day) was not implemented. All participants underwent nutritional education and training and were provided the protocol. During the study, they received daily reminders such as phone calls and text messages to confirm and encourage them to eat according to a specific protocol.

#### 2.4.2. Physical Examination and Blood Test

All participants underwent anthropometric and blood tests at baseline and after the 2-week intervention period. Height (Seca 264, Seca GmbH & Co. KG, Hamburg, Germany) and weight (Inbody 770; Biospace Co., Ltd., Seoul, Republic of Korea) were measured to the first decimal place using a digital scale in the fasting state. BMI was calculated as weight (kg) divided by height (m^2^). The participants in this study were men (*n* = 13) and women (*n* = 15) between the ages of 20 and 65 years with a BMI of 25 kg/m^2^ or more. Participants completed a health questionnaire as a screening tool. Total body fat (%), visceral fat area (cm^2^), and lean body mass (kg) were measured using a bioelectric impedance analyzer (InBody 770; Biospace Co., Ltd., Seoul, Republic of Korea). Resting blood pressure and heart rate were measured after a 10 min rest, and brachial systolic blood pressure (SBP), diastolic blood pressure (DBP), and heart rate were measured using a digital blood pressure monitor.

Blood was collected from all subjects on the morning of the test day after an 8 h fast. Fasting blood glucose, total cholesterol, triglycerides, high-density lipoprotein (HDL) cholesterol, and low-density lipoprotein (LDL) cholesterol levels were measured using a Hitachi Automatic Analyzer 7600 (Hitachi Ltd., Tokyo, Japan). Serum insulin concentrations were estimated using an immunoradiometric assay (BioSource, Nivelles, Belgium). The homeostasis model assessment of insulin resistance (HOMA-IR) was calculated using the following equation: [fasting insulin level (IU/mL) × fasting glucose level (mg/dL)]/405. The levels of advanced glycation end products (AGEs) in the skin were estimated using skin autofluorescence measurements recorded using an AGE Reader (DiagnOptics Technologies B.V., Groningen, The Netherlands). AGEs were measured using a skin autofluorescence reader (for instance, the AGE Reader mu Connect; DiagnOptics), because, although tissue biopsy is ideal, it is also invasive. Skin autofluorescence is measured using an AGE reader, which illuminates a small portion of the skin (approximately 4 cm^2^) on the inside of the forearm of the subject. These values are computerized and generated in arbitrary units via a series of calculations. The measured AGE values correlated well with the specific AGE values, including pentosidine, carboxymethyl-lysine, and carboxyethyl lysine values in skin biopsy readings [24]. Three consecutive measurements of skin autofluorescence were performed on the inner side of the dominant forearm, and the mean of the three measurements was used for analysis.

#### 2.4.3. Oral Glucose Tolerance Test (OGTT)

To determine the effect of Dodamssal on glucose metabolism, an OGTT was performed on 10 of the study subjects who agreed to participate in it. Blood samples were taken before the participants were served Dodamssal or control rice. After consuming the food of their respective groups, the subjects had blood samples collected every 30 min until 2 h had elapsed.

### 2.5. Statistical Analysis

All data are presented as the mean and standard deviation of triplicate measurements, and were analyzed using SAS v. 9.2 (SAS Institute Inc., Cary, NC, USA). Statistical significance was determined using one-way analysis of variance and Duncan’s multiple comparisons. *p* < 0.05 indicated statistically significant differences between treatment means.

## 3. Results and Discussion

### 3.1. Nutritional Component Analysis

Table 2 shows the nutritional component analysis results of the meal samples used in the clinical trials, and that of HBI and HBD, which comprised their main ingredient, respectively. The meals were prepared considering the different caloric content of HBI and HBD, as described in Section 2.1. However, there was a statistically significant difference in calories owing to the higher calorie content in the HBD meal than in the HBI meal, which is believed to be due to the difference in fat content. Compared to their respective HBI and HBD powders, the protein and fat contents of the HBI and HBD meals were higher, and the dietary fiber and carbohydrate contents were lower. In addition, dietary fiber, carbohydrate, and protein contents showed no statistical difference between the HBI and HBD meals.

### 3.2. Particle Size Distribution of Heat-Treated Brown Rice Flours and Powdered Meals

The particle size distribution and median diameter of particles in the heat-treated brown rice flours and meals used in the clinical trials are shown in Table 3. In the smallest particle size distribution range (0–20 μm), the highest percentage was observed for HBD (24.2 ± 0.7%) and the lowest for HBI (18.6 ± 0.6%). However, no significant differences were observed between the HBI and HBD groups. The median particle diameter ranged from 52% to 68% in the following order: HBI meals > HBI > HBD meals > HBD. There were no statistical differences among all samples.

In previous studies, the particle size of rice powder was in the order of high-amylose rice, low-amylose rice, and medium-amylose rice, from largest to smallest, and the high-amylose rice showed the largest starch particles, even in separated starch [9]. In general, the content of amylose and RS is positively correlated, and the particle size decreases as the amylose content increases in corn starch [25]. In barley, RS exhibited the highest particle size range of 1–15 μm [26]. In the current study, HBD exhibited the highest percentage among the samples in the range of 0–20 μm, which includes the smallest particles, and the sample containing RS exhibited a small median diameter, in contrast to the previous results reported on rice [9].

### 3.3. RS Content and In Vitro Starch Digestibility

Figure 1 illustrates the RS content, in vitro starch digestibility, and eGIs of the various samples. The RS content in HBD powder was 13.8 ± 0.1%, and in its associated meal (HBD), it was 11.4 ± 0.1%, which was slightly reduced (Figure 1a). However, there was no difference in RS content between HBI and its associated meal (HBI), which contained almost no RS (<1%). In terms of in vitro digestibility, HBI and HBI meal samples showed higher digestibility than that of Dodamssal-containing samples (HBD and HBD meals). HBI and HBD meals, which had reduced carbohydrate content, showed lower starch digestibility than their respective HBI and HBD powders (Figure 1b). Low digestibility showed consistent results with low HI and eGI results. The eGI of HBI and HBD were 70.5 ± 0.5 and 56 ± 0.3, respectively, showing a statistical difference among all samples (Figure 1c).

Factors that affect starch digestibility include physical encapsulation, starch composition and structure, presence of proteins and lipids, food processing and cooking, and the addition of functional ingredients [6]. In previous studies, Dodamssal with a high amylose content of 40%, a high ratio of long-chain amylose to amylopectin, and high C-type starch crystallinity had a lower RDS and higher SDS and RS content than that of A-type rice starch [11]. In the current study, although heat-treated brown rice samples contained lipids and proteins, consistent results were found for HBD meals containing Dodamssal in terms of high RS content and low digestibility.

As shown in Table 3, the smaller the median particle diameter, the higher the distribution ratio of small particles (0–20 μm). Additionally, the lower the distribution of large particles (101–400 μm), the higher the RS content and the lower the starch digestibility, thus resulting in lower HI and eGI (Figure 1).

Zhu et al. [9] reported that there was no correlation between rice flour, rice starch particle size, and digestibility, but Qi and Tester [27] reported that small starch particles, except for high-amylose starch, were quickly digested by enzymes. In the current study, the results were inconsistent with the finding that small starch particle size is generally associated with high enzymatic hydrolysis since high-amylose rice was used as the test material.

### 3.4. Clinical Study

#### 3.4.1. Participant Characteristics

The mean ages of the test group (TG) and control group (CG) were similar at 34.93 ± 0.69 years and 35.79 ± 0.72 years, respectively. Each participant was randomized according to a crossover design. There were no significant differences between the two groups in the other physical examinations, including anthropometric measurements (Table 1). After 2 weeks of ingesting the prepared meals, body weight, total body fat, and WC decreased in both groups; however, there were no significant differences (Table 4). It is thought that the 2-week intervention was too short to induce changes in body composition, including body weight. RS is the portion of starch that is not digested and absorbed in the small intestine; it acts as a mild laxative because it plays a role similar to that of dietary fiber in the intestines [28]. Fiber intake through RS increases satiety and affects weight loss because satiety hormones such as glucagon-like peptide-1 and peptide YY are secreted [29]. However, the 2-week intervention was too short for RS to reduce appetite and decrease body fat.

#### 3.4.2. Effects of Glycemic Control

Outcomes of OGTT Glycemic Control

The OGTT, which measures the body’s response to glucose, is used to screen for or diagnose T2DM and gestational diabetes, a type of diabetes that develops during pregnancy. The OGTT identifies abnormalities in the way the body regulates glucose levels after a meal or often before fasting blood sugar levels become abnormal. There were no statistically significant differences in the results of the OGTT, even after 2-week intervention, between the two participant groups (Figure 2). This observation may be due to the short duration of the intervention and the small number of participants. As shown in Table 4, fasting blood glucose decreased by 0.50 ± 1.98 mg/dL in the CG and 1.64 ± 3.01 mg/dL in the TG (*p* = 0.345). The changes in glycated hemoglobin (HbA1c), which reflects 2–3 months of blood glycemic control, increased by 0.01 ± 0.11% in the CG and decreased by 0.04 ± 0.11% in the TG after 2 weeks, but there was no significant difference between the two groups (*p* = 0.221). Nevertheless, glycemic metabolism tended to improve in the TG group, especially since fasting insulin secretion decreased significantly by 5.71 ± 5.3 uU/dL in the TG when compared to 2.21 ± 0.58 uU/dL in the CG. For this reason, it was confirmed that HOMA-IR, an insulin resistance index, decreased significantly in the TG. There was also a significant difference between the two groups in AGE accumulation in the subcutaneous tissue inside the skin of the upper arm, which decreased by 2.99% in the TG and increased by 7.45% in the CG (*p* = 0.003). We hypothesized that RS would have beneficial effects on blood glucose metabolism, in addition to regulating gut microbiota, satiety hormones, and gastric emptying [25]. The only significant difference in intake between the two groups was the amount of RS; thus, it is likely that RS had a major effect on glucose metabolism. We found that the HBD group, which consumed Dodamssal, had a much higher RS content and much lower eGI than those of the HBI group, which consumed Ilmi (Figure 1).

2.Previous RS Studies

Nevertheless, the results of studies on the effects of RS on glycemic control are controversial. Several systematic reviews and meta-analyses have reported the metabolic and anti-inflammatory effects of RS [30,31,32,33,34,35]. Some of these studies have shown positive effects of RS on the regulation of glucose metabolism in patients with diabetes. RS has also shown positive effects in regulating glucose metabolism, such as lowering fasting glucose levels and improving insulin resistance in patients with diabetes [32,34]. However, most clinical studies have involved small sample sizes (12–75 patients) and short intervention periods (2–12 weeks). Another important issue to consider is the heterogeneity of the RS and the composition of the gut microbiome [28]. In fact, there are differences in the RS type and content selected for each study, and metabolic reactions in the intestine differ depending on the composition of the gut microbiome [34]. Finally, dietary intake may vary among individuals, leading to changes in insulin secretion, glucose homeostasis, and lipid levels. In addition to the advantages of RS against insulin resistance, owing to its anti-inflammatory effect and improvement of the gut microbiome, there have been systematic reviews related to patients with chronic kidney disease (CKD) or inflammatory bowel disease (IBD) [30,33]. RS significantly improved the antioxidant capacity and reduced blood C-reactive protein levels in patients with T2DM and in patients with CKD [30,34]. Recently, it was reported that RS is associated with improved clinical remission in patients with IBD and improved gut microbiome imbalance in patients with chronic kidney disease [30,33].

3.Strengths of this Study

Unlike previous studies using corn extraction [36], the present study evaluated the function of RS extracted from Dodamssal rice. Furthermore, it is meaningful to confirm that RS has a beneficial effect, even in healthy obese participants without other diseases, such as T2DM or CKD. It is also important that this study administered RS in the form of a powdered food rather than a drug, such as a capsule [37].

#### 3.4.3. Effects of Lipid Control

LDL cholesterol decreased by 5.14 ± 2.53 mg/dL in the HBD group and by 8.29 ± 4.15 mg/dL in the HBI group after 2 weeks of dietary intake, but the difference was not significant (*p* = 0.451). In addition to LDL, lipid levels such as triglycerides and total cholesterol improved in both groups over time, but there was no significant difference between the two groups. As described above, lowering blood glucose levels by supplying RS causes obvious changes in glucose metabolism by promoting glycogen synthesis and inhibiting gluconeogenesis. Moreover, consuming RS promotes the growth of short-chain fatty acid (SCFA)-producing bacteria. Increases in SCFA production are expected to lead to a decrease in intestinal pH, thereby reducing pro-inflammatory and pro-oxidative stress responses [34]. However, the intervention period was too short to confirm an improvement in lipid metabolism through the promotion of lipid oxidation and cholesterol homeostasis [28,30]. As RS favorably affects glucose metabolism, long-term RS supplementation may affect body weight control and body composition.

#### 3.4.4. Other Laboratory Findings

Aspartate aminotransferase (AST) and alanine aminotransferase (ALT) levels increased slightly over the course of the trial. AST increased by 2.07 ± 2.84 mg/dL in the HBD group and by 1.0 ± 1.2 mg/dL in the HBI group, while ALT increased by 2.64 ± 1.30 mg/dL in the HBD group and by 2.86 ± 4.22 mg/dL in the HBI group; there were no significant differences between the two groups (Table 4). These changes were slight and may have been temporary and insignificant. One possible reason is that the study period was quite short, and the possibility of a slight increase is owed to a rapid change in eating pattern over 2 weeks. In addition, changes in other liver parameters, such as gamma glutamyltransferase and bilirubin levels, were unclear; therefore, they are not clinically meaningful.

#### 3.4.5. Tolerability Analysis

We expected that the reduction in colonic transit time following RS intake would trap water in the intestine and prevent stool from drying out, ultimately preventing constipation [38]. However, its effect on constipation prevention has not been confirmed. Of the thirty-three participants, five dropped out of the study. However, none of the subjects dropped out owing to side effects of the intervention, such as abdominal discomfort, flatulence, diarrhea, constipation, nausea, and fullness. One out of five participants who dropped out from the clinical trial had a compliance of 59%, and complained of difficulty in eating only the delivered food. However, this participant did not complain of digestive side effects or pruritus owing to the powder intervention. One participant took a contraindicated drug, and the other three withdrew their consent and were excluded for personal reasons. Ultimately, 28 participants (14 each in the HBI and HBD groups) were included in the final analyses. The compliance rate of the final 28 patients was 84%, and there were no adverse side effects reported. This may be because our study used RS directly as a food in powder form, when compared to other studies that used capsules or tablets. There was only one complaint about the inconvenience of dissolving the powder in water. Otherwise, we cannot rule out the possibility that the underlying dietary variability in noncompliance with dietary recommendations during the intervention period may have had an impact. In other words, even with similar compliance reports from participants, unreported alterations in dietary patterns cannot be disregarded. Nevertheless, according to the results of a rat study, the use of RS can improve intestinal health and, subsequently, reduce the incidence of post-weaning diarrhea and associated mortality [39]. Therefore, further large-scale clinical studies with larger sample sizes are required.

## 4. Conclusions

We conducted clinical trials to evaluate Dodamssal, which has an increased RS content after heat treatment, as a healthy food additive in powdered meals. HBD powder exhibited a high RS content, small median particle diameter, and a low eGI; it was used as a raw material for HBD meals. The HBD group with RS in the meals and the HBI group without RS were supplied for 2 weeks with other foods and powdered test materials. Although there were no significant differences in the diet results comparing body weight, body composition, cholesterol levels, and OGTT measurements, even after 2 weeks of intervention, increased RS consumption showed a statistically significant positive effect on glucose metabolism. The changes in HbA1c, which reflect 2–3 months of blood glycemic control, increased by 0.01 ± 0.11% in the HBI group and decreased by 0.04 ± 0.11% in the HBD group after 2 weeks, but the difference between the groups was not significant (*p* = 0.221). RS supplementation for 2 weeks had a beneficial effect on glucose control in obese participants. Large-scale studies are needed to determine whether these results are influenced by differences in the gut microbiome, underlying dietary variability, or other environmental factors.

## Figures and Tables

**Figure 1 nutrients-15-02248-f001:**
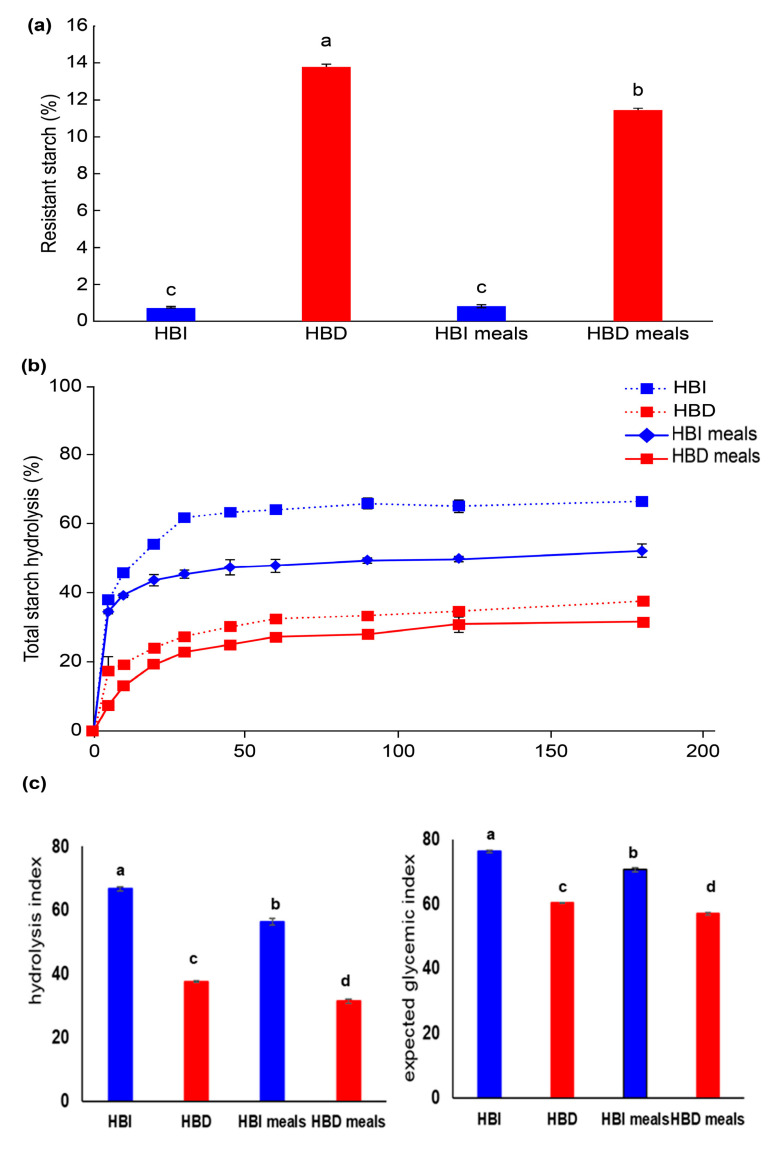
Starch digestibility of heat-treated brown rice flours (HBI and HBD) and tested sample meals (HBI and HBD meals). (**a**) Resistant starch contents, (**b**) total starch hydrolysis, and (**c**) hydrolysis index and expected glycemic index in heat-treated brown rice flours and sample meals. Values with different letters above the bars are significantly different (*p* < 0.05), as determined by Duncan’s multiple range test.

**Figure 2 nutrients-15-02248-f002:**
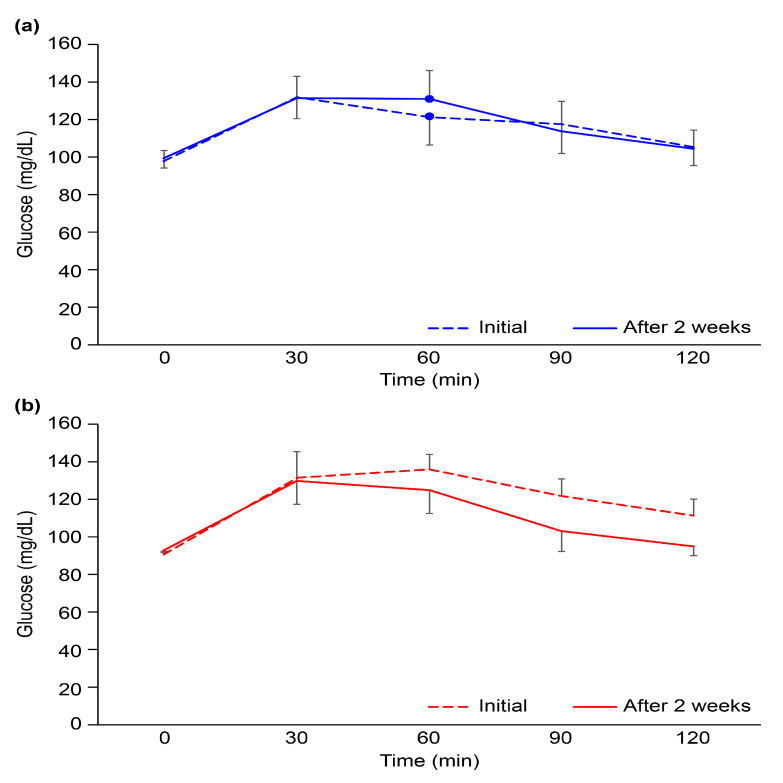
Oral glucose tolerance test results at the initiation of the clinical trial and after 2 weeks in the (**a**) heat-treated brown rice Ilmi and (**b**) heat-treated brown rice Dodamssal groups. Black bars represent the standard deviation.

**Table 1 nutrients-15-02248-t001:** Baseline characteristics of all participants.

	HBI Group	HBD Group	*p*-Value
Weight, kg	80.79 ± 2.70	81.07 ± 3.04	0.945
BMI, kg/m^2^	27.93 ± 0.68	28.64 ± 0.70	0.469
WC, cm	93.57 ± 2.36	94.21 ± 1.80	0.830
Total body fat, %	33.64 ± 1.85	33.29 ± 2.21	0.902
Visceral fat area, cm^2^	123.43 ± 8.67	123.21 ± 10.43	0.988
Lean body mass, kg	30.36 ± 1.58	30.36 ± 1.62	1.000
SBP, mmHg	126.21 ± 2.34	126.0 ± 1.74	0.942
DBP, mmHg	77.79 ± 1.48	76.71 ± 1.43	0.607
Pulse, rate/min	75.43 ± 2.52	71.36 ± 1.77	0.199
Fasting glucose, mg/dL	99.21 ± 2.32	101.14 ± 2.28	0.558
Plasma insulin, µU/dL	12.50 ± 1.77	15.21 ± 2.08	0.329
HbA1c, %	5.50 ± 0.14	5.71 ± 0.13	0.262
HOMA-IR	3.07 ± 0.47	3.93 ± 0.59	0.267
AGE	1.88 ± 0.26	2.01 ± 0.32	0.229
Triglyceride, mg/dL	134.50 ± 23.14	132.21 ± 19.57	0.940
Total cholesterol, mg/dL	193.36 ± 8.87	198.57 ± 6.68	0.643
LDL cholesterol, mg/dL	115.57 ± 8.13	114.43 ± 4.97	0.906
HDL cholesterol, mg/dL	55.79 ± 3.61	57.71 ± 4.33	0.735
Apo A1, mg/dL	164.71 ± 5.50	167.43 ± 8.45	0.790
Apo B, mg/dL	96.93 ± 5.98	100.14 ± 4.86	0.680
Uric acid, mg/dL	5.93 ± 0.32	6.14 ± 0.42	0.688
AST, U/L	19.07 ± 1.43	21.50 ± 2.42	0.398
ALT, U/L	23.43 ± 3.49	30.86 ± 6.81	0.344
rGTP, U/L	24.50 ± 2.74	28.71 ± 6.36	0.548
BUN, mg/dL	11.64 ± 0.59	13.07 ± 0.74	0.143
Creatinine, mg/dL	0.85 ± 0.09	0.86 ± 0.10	1.000
eGFR, mL/min/1.73 m^2^	109.86 ± 6.21	105.64 ± 5.67	0.621

Values are presented as means ± SD of all participants. AGE: advanced glycation end-product, Apo A1: apolipoprotein AI, Apo B: apolipoprotein B, ALT: alanine aminotransferase, AST: aspartate aminotransferase, BMI: body mass index; DBP: diastolic blood pressure; eGI: estimated glycemic index, HbA1c: glycated hemoglobin, HBI: heat treated brown rice Ilmi, HBD: heat treated brown rice Dodamssal, HDL: high-density lipoprotein, HOMA-IR: homeostasis model assessment for insulin resistance, LDL: low-density lipoprotein, rGTP: gamma-glutamyltransferase, SBP: systolic blood pressure, WC: waist circumference.

**Table 2 nutrients-15-02248-t002:** Nutritional value of heat-treated brown rice flours (HBI and HBD) and tested sample meals (HBI and HBD).

Samples	Energy (kcal/100 g)	Protein (g/100 g)	Fat (g/100 g)	Dietary Fiber (g/100 g)	Carbohydrate (g/100 g)
HBI	393.0 ± 1.1 ^c^	8.0 ± 0.2 ^b^	3.4 ± 0.1 ^c^	4.1 ± 0.1 ^b^	82.7 ± 0.5 ^b^
HBD	405.0 ± 1.3 ^a^	7.8 ± 0.3 ^b^	3.3 ± 0.1 ^c^	4.6 ± 0.1 ^a^	86.0 ± 0.6 ^a^
HBI meals	395.4 ± 0.7 ^c^	9.4 ± 0.3 ^a^	6.4 ± 0.2 ^b^	2.7 ± 0.1 ^c^	76.3 ± 0.6 ^c^
HBD meals	399.2 ± 3.1 ^b^	9.9 ± 0.6 ^a^	7.0 ± 0.1 ^a^	2.8 ± 0.1 ^c^	75.5 ± 0.7 ^c^

^a–c^ Values with different letters within a column are significantly different (*p* < 0.05), as determined by Duncan’s multiple range test. Values are presented as means ± SD of three independent trials. HBI: Heat-treated brown rice of ‘Ilmi’, HBD: heat-treated brown rice of ‘Dodamssal’, HBI meals: control sample meal containing HBI, HBD meals: test sample meal containing HBD.

**Table 3 nutrients-15-02248-t003:** Particle size distribution and median diameter of particles in heat-treated brown rice flours and tested meals.

Samples	Particle Size Distribution (%)						Median Diameter (μm)
	0–20 (μm)	20–40 (μm)	40–60 (μm)	60–100 (μm)	100–200 (μm)	200–400 (μm)	
HBI	18.6 ± 0.6 ^c^	16.3 ± 0.5 ^ns^	11.6 ± 0.3 ^b^	22.8 ± 0.2 ^b^	23.3 ± 0.7 ^a^	7.1 ± 0.6 ^a^	68.2 ± 0.8 ^b^
HBD	24.2 ± 0.7 ^a^	20.5 ± 0.7 ^ns^	14.3 ± 0.3 ^a^	24.5 ± 0.2 ^a^	16.3 ± 1.2 ^b^	0.5 ± 0.2 ^c^	52.0 ± 0.3 ^d^
HBI meals	21.1 ± 0.6 ^b^	16.6 ± 0.5 ^ns^	11.9 ± 0.3 ^b^	24.1 ± 0.3 ^ab^	23.7 ± 1.0 ^a^	2.8 ± 0.7 ^b^	74.1 ± 0.4 ^a^
HBD meals	21.6 ± 0.6 ^b^	17.8 ± 0.6 ^ns^	12.7 ± 0.3 ^b^	24.2 ± 0.2 ^ab^	21.9 ± 1.0 ^a^	2.1 ± 0.6 ^bc^	60.2 ± 0.9 ^c^

^a–d^ Values with different letters within the same column are significantly different (*p* < 0.05), as determined by Duncan’s multiple range test. ^ns^: not statistically significant. Values are presented as means ± SD of three independent trials. HBI: Heat-treated brown rice of ‘Ilmi’, HBD: heat-treated brown rice of ‘Dodamssal’, HBI meals: control sample meal containing HBI, HBD meals: test sample meal containing HBD.

**Table 4 nutrients-15-02248-t004:** Differences in participant measurements between the initiation of the study and after 2-week human intervention in the control vs. test group.

	HBI Group (%)	HBD Group (%)	*p*-Value
Weight, kg	−2.86 ± 0.69 (3.54)	−2.29 ± 0.41 (2.82)	0.070
WC, cm	−2.86 ± 0.69 (3.06)	−4.14 ± 0.66 (4.40)	0.872
Total body fat, %	−1.14 ± 0.29 (3.39)	−1.01 ± 0.35 (3.03)	0.905
Visceral fat area, cm^2^	−5.64 ± 1.20 (2.14)	−5.86 ± 1.20 (4.76)	0.114
Lean body mass, kg	−1.43 ± 0.25 (4.71)	−1.43 ± 0.77 (4.71)	0.937
Fasting glucose, mg/dL	−0.50 ± 1.98 (0.50)	−1.64 ± 3.01 (1.62)	0.345
Plasma insulin, uU/dL	−2.21 ± 0.58 (17.68)	−5.71 ± 5.30 (37.54)	0.021
HbA1c, %	0.01 ± 0.11 (0.18)	−0.04 ± 0.11 (0.70)	0.221
HOMA-IR	−0.50 ± 0.14 (16.29)	−1.50 ± 1.40 (38.17)	0.023
AGEs, AU	0.14 ± 0.18 (7.45)	−0.06 ± 0.14 (2.99)	0.003
Triglyceride, mg/dL	−22.86 ± 11.30 (17.0)	−17.86 ± 12.78 (13.51)	0.772
Total cholesterol, mg/dL	−20.0 ± 4.3 (10.34)	−26.57 ± 6.22 (13.38)	0.395
LDL cholesterol, mg/dL	−5.14 ± 2.53 (4.45)	−8.29 ± 4.15 (7.24)	0.451
HDL cholesterol, mg/dL	−8.50 ± 1.98 (51.23)	−10.79 ± 2.25 (18.70)	0.524
Apo A1, mg/dL	−16.64 ± 3.43 (10.10)	−26.07 ± 4.22 (15.57)	0.095
Apo B, mg/dL	0.93 ± 2.40 (0.96)	−4.57 ± 4.00 (4.56)	0.252
AST, U/L	1.0 ± 1.2 (5.24)	2.07 ± 2.84 (9.63)	0.553
rGTP, U/L	−5.14 ± 1.98 (20.98)	−7.57 ± 2.41 (26.37)	0.443
Uric acid, mg/dL	−0.07 ± 0.19 (1.18)	−0.43 ± 0.17 (7.0)	0.127

AGEs: advanced glycation end products, Apo A1: Apolipoprotein AI, Apo B: Apolipoprotein B, ALT: alanine aminotransferase, AST: aspartate aminotransferase, BMI: body mass index, HBI: heat-treated brown rice ‘Ilmi’, HBD: heat-treated brown rice ‘Dodamssal’, HbA1c: glycated hemoglobin, HDL: high-density lipoprotein, HOMA-IR: homeostasis model assessment for insulin resistance, AU: arbitrary units; LDL: low-density lipoprotein, rGTP: gamma-glutamyltransferase, WC: waist circumference.

## Data Availability

The data presented in this study are available on request from the corresponding author.

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
