# Peer review of "Effects of Consuming Heat-Treated Dodamssal Brown Rice Containing Resistant Starch on Glucose Metabolism in Humans"

_nutrients, 2023, doi:10.3390/nu15102248_

Round 1
Reviewer 1 Report
The presented paper presents interesting research results, it is clear that the authors have prepared it with great care. Nonetheless, I have a few questions and recommendations that should be included in the article to improve its merit. Here are my recommendations:
1. What are the potential benefits of resistant starch and how can it be enhanced in rice cultivars?
2. Please add a paragraph to the introduction in which the authors describe what will be presented in the article. Also, describe what research was conducted.
3. I think this topic very interesting, but I propose to do a better literature review and supplement the introduction with additional literature items.
4. Statistics in graphs should be applied using the symbol - *- this will improve the readability of the article.
5. Please elaborate on the discussion and present it in bulleted form.
The text is written linguistically correct with great care. Please note only minor editorial errors such as in line 64 : ...'Dodamssal,' with.... etc.
Author Response
Reviewer 1
Comments and Suggestions for Authors
The presented paper presents interesting research results, it is clear that the authors have prepared it with great care. Nonetheless, I have a few questions and recommendations that should be included in the article to improve its merit. Here are my recommendations:
→ We appreciate your review of our manuscript. We have revised the manuscript according to the comments from the reviewers and have sought professional help for English language editing. We have responded to each of the comments and have attached the editing certificate with this submission.
- What are the potential benefits of resistant starch and how can it be enhanced in rice cultivars?
→ We added the potential effects of RS to the revised manuscript (lines 47–49). We also added the background of the development of Dodamssal to the revised manuscript (lines 67–71). To generate resistant starch in the breeding stage, Goami 2 rice with a B-type crystal structure was developed using N-methyl-N-nitrosourea treatment. In addition, there are several processes that increase RS in food technology. Representatively, it is retrogradation that occurs after heat processing (our manuscript also presents previous studies related to this in lines 78–80).
- Please add a paragraph to the introduction in which the authors describe what will be presented in the article. Also, describe what research was conducted.
→ We have described some prior research and purpose in our original manuscript, but we added additional content to the introduction section according to the reviewer's suggestion. The content and background of the research that we conducted are described in lines 71–80.
- I think this topic very interesting, but I propose to do a better literature review and supplement the introduction with additional literature items.
→ Thank you for your comment. We revised and supplemented the introduction. However, in consideration of another reviewer's suggestion to reduce the references by half, we included mainly information from our previous research (red font in the introduction section).
- Statistics in graphs should be applied using the symbol - *- this will improve the readability of the article.
→ Thank you pointing this out. We plotted all graphs in alphabetical terms concerning statistical significance. We also added letters on the bars of Fig. 1(c) that were missing. However, since we performed Duncan’s multiple test, we classified it with alphabet letters and not the asterisk symbol “*,” which is mainly displayed in the t test.
- Please elaborate on the discussion and present it in bulleted form.
→ According to your suggestion, we have presented some of the discussion using bullets.
Comments on the Quality of English Language
The text is written linguistically correct with great care. Please note only minor editorial errors such as in line 64 : ...'Dodamssal,' with.... etc.
→ Thank you for your comment. We revised “…'Dodamssal,' with…” to “…Dodamssal, which has…” in the manuscript, and revised “…‘Dodamssal’…” to “…Dodamssal…” (lines 67, 79, 101, 260, and 283).

Reviewer 2 Report
This manuscript describes the effects of co sumie brown rice wirh resistant starych on the metabolism of glucose. Clinical trial meals were prepared for 36 obese participants. The Authors concluded that supplementation of resistant starch for 2 weeks appears to have beneficial effect on glycemic control in obese participants. In my opinion the manuscript is well written and the methodology is correct. Before the publucation in Nutrients I will sugggest to refresh the references as only half of them was published in the last five years.
Author Response
Comments and Suggestions for Authors
This manuscript describes the effects of co sumie brown rice with resistant starch on the metabolism of glucose. Clinical trial meals were prepared for 36 obese participants. The Authors concluded that supplementation of resistant starch for 2 weeks appears to have beneficial effect on glycemic control in obese participants. In my opinion the manuscript is well written and the methodology is correct. Before the publication in Nutrients I will suggest to refresh the references as only half of them was published in the last five years.
→ We appreciate your review of our manuscript. Following your suggestion, we deleted five references and added the latest reference. However, we added four of our prior studies in response to another reviewer's opinion to supplement the literature in the introduction. Therefore, we hope that you would understand that although many references have been deleted with the intent of reducing the list in half, this inevitably could not be achieved.
Deleted references: (initial number of references)
1.Rössner, S. Obesity: the disease of the twenty-first century. Int J Obes Relat Metab Disord 2002, 26, Suppl 4, S2–4. DOI:10.1038/sj.ijo.0802209.
2.Adam-Perrot, A.; Clifton, P.; Brouns, F. Low-carbohydrate diets: nutritional and physiological aspects. Obes Rev 2006, 7, 49–58. DOI:10.1111/j.1467-789X.2006.00222.x.
- Jeong, O.Y.; Park, H.-S.; Baek, M.-K.; Kim, W.-J.; Lee, G.-M.; Lee, C.-M.; Bombay, M.; Ancheta, M.B.; Lee, J.-H.; Ancheta, M.B.; Lee, J. Review of rice in Korea: current status, future prospects, and comparisons with rice in other countries. J Crop Sci Biotechnol 2021, 24, 1–11. DOI:10.1007/s12892-020-00053-6.
- Franssila-Kallunki, A.; Rissanen, A.; Ekstrand, A.; Ollus, A.; Groop, L. Weight loss by very-low-calorie diets: effects on substrate oxidation, energy expenditure, and insulin sensitivity in obese subjects. Am J Clin Nutr 1992, 56(1), Suppl, 247S–248S. DOI:10.1093/ajcn/56.1.247S.
- Grabitske, H.A.; Slavin, J.L. Gastrointestinal effects of low-digestible carbohydrates. Crit Rev Food Sci Nutr 2009, 49, 327–360. DOI:10.1080/10408390802067126.

Reviewer 3 Report
The investigation entitled “Effects of consuming heat-treated Dodamssal brown rice-containing resistant starch on glucose metabolism in humans” exposes the results obtained during a clinical study directed to investigate the impact of diet including brown rice of different cultivars.
In the actual context of the increasing population diagnosed with obesity, it is vital to develop new strategies able to ensure healthier diet.
Overall, the paper is well written, the cited references are pertinent, the applied methods are well described, the results are presented with enough clarity and the conclusions sustain the data.
At the same time though, it is recommended to include answers to the following questions:
1. Why choosing these particular rice cultivars?
2. Are there results with which the reported ones for these cultivars can be compared?
3. Why choosing the heat treatment method for rice preparation?
Only minor editing of English language is required.
Author Response
Comments and Suggestions for Authors
The investigation entitled “Effects of consuming heat-treated Dodamssal brown rice-containing resistant starch on glucose metabolism in humans” exposes the results obtained during a clinical study directed to investigate the impact of diet including brown rice of different cultivars.
In the actual context of the increasing population diagnosed with obesity, it is vital to develop new strategies able to ensure healthier diet.
Overall, the paper is well written, the cited references are pertinent, the applied methods are well described, the results are presented with enough clarity and the conclusions sustain the data.
At the same time though, it is recommended to include answers to the following questions:
→ We appreciate your review of our manuscript. We have revised the manuscript according to the comments from the reviewers and have sought professional help for English language editing. We have responded to each of the comments.
- Why choosing these particular rice cultivars?
→ The rice cultivar used for this study (Dodamssal) have been newly developed and one of the first C-type rice developed through Korean domestic conventional breeding methods. In the early 2000s in Korea, when Goami-2 (rice cultivar) exhibiting B-type crystallinity was developed, researchers from China showed increased interest, which led to further research efforts on developing high-amylose breeds. The development of C-type rice was a culmination of these efforts. However, since it was developed via genetic modification instead of conventional breeding methods, it was not readily accepted by the food industry. Subsequent research work led to Goami-2, -3, and -4 types (cultivars) being developed as B-type starch cultivars. Finally, Dodamssal was developed, which exhibits significant differences from the three previously developed breeds. Dodamssal is a Japonica-type high-amylose breed developed using traditional breeding from Goami (a breed with an amylose content of 26.7%) and Goami2 (a mutant breed developed using N-methyl-N-nitrosourea (MNU) treatment and containing resistant starch and about 33% of amylose (Lines 67-71 in our manuscript). Dodamssal had the lowest digestibility, highest RS content, and showed potential for use as a source of starch for weight loss and hypoglycemic effects owing to its low glycemic index. We hope that this study will be a valuable for future research and development efforts in the food industry.
- Are there results with which the reported ones for these cultivars can be compared?
→ We have described some prior research in our original manuscript (Lines 71-80).
References are provided below.
- Park, J.; Oh, S.-K.; Chung, H.-J.; Park, H.-J. Structural and physicochemical properties of native starches and non-digestible starch residues from Korean rice cultivars with different amylose contents. Food Hydrocoll. 2020, 102, 105544. DOI:1016/j.foodhyd.2019.105544.
→ In this study, rice starches from four cultivars: Baegokchal (BOC), Ilmi (IM), Mimyeon (MM), and Dodamssal (DDS), were studied in terms of their physicochemical and structural features. Native starches (NS) from MM and DDS showed high amylose content and low rapidly digestible starch, as well as high slowly digestible starch and resistant starch (RS) ratios. To elucidate the characteristics of RS in rice, non-digestible starches (NDS) were isolated from NS from each cultivar. The starch crystallinity of BOC, IM, and MM showed an A-type X-ray diffractometry pattern; however, DDS granules displayed a C-type crystallinity pattern with a predominant B-type. DDS starch granules had a convex spherical shape, whereas BOC, IM, and MM starch granules had a polygonal shape. All starches from IM and BOC were hydrolyzed, with no NDS residues remaining. The NDS from MM and DDS, which are high-amylose cultivars, showed a lower molecular weight, longer average amylopectin chain length, and lower viscosity than NS. DDS had the lowest digestibility, highest RS content, and showed potential for use as a source of starch for weight loss and hypoglycemic effects owing to its low glycemic index. The low viscosity of DDS can potentially be exploited for its use as a daily dietary component through the development of suitable processing methods for products such as rice noodles.
- Park, J.; Oh, S.-K.; Chung, H.-J.; Shin, D.S.; Choi, I.; Park, H.-J. Effect of steaming and roasting on the quality and resistant starch of brown rice flour with high amylose content. LWT. 2022, 167, 113801. DOI:1016/j.lwt.2022.113801.
→ The high-amylose rice cultivar Dodamssal (DDS), a potentially nutritious functional food, contains resistant starch (RS). However, its RS content varies with the processing method used. In this study, heat treatment was used to produce rice powder with enhanced RS content. The rice powder quality was improved by steaming rough rice and brown rice for 30 min and roasting the brown rice at 240 ℃ for 10 min. Brown rice flour (BRF) made from steamed rough rice and roasted DDS (SRRD) had higher RS content (14%), ratio of fine particle sizes (0–20 µm), and gelatinization degree (GD), as well as lower mean particle size (51 µm) and in vitro glycemic index compared to that of roasted DDS. Contrastingly, BRF prepared from steamed brown rice and roasted DDS with high GD had low RS content (12%) and high mean particle size (56 µm). Hence, SRRD-BRF improves powder quality and may be beneficial for nutritious functional food production.
- Rivera-Piza, A.; Choi, L.; Seo, J.; Lee, H.G.; Park, J.; Han, S.I.; Lee, S.J. Effects of high-fiber rice Dodamssal (Oryza sativa L.) on glucose and lipid metabolism in mice fed a high-fat diet. J Food Biochem. 2020, 44, e13231. DOI:10.1111/jfbc.13231.
→ We investigated the effects of high amylose rice variety, Dodamssal (DO) (Oryza sativa L.), on glucose homeostasis and lipid metabolism in mice. Experiment 1: Oral administration of DO for 1 week significantly improved glucose and insulin tolerance (p < .001) and reduced plasma triglyceride and low-density lipoprotein cholesterol concentrations. Experiment 2: Administration of DO-containing diet for 5 weeks also significantly reduced fasting glucose concentrations and hepatic lipid accumulation. DO induced GLP-1, adiponectin, and PYY levels. In the liver, DO suppressed the gene expression of G6pc, key gene in gluconeogenesis and induced AKT phosphorylation. DO increased fecal bile acid excretion regulating the expression in key genes in bile acid metabolism. DO suppressed plasma Trimethylamine N-oxide and intestinal lipopolysaccharide concentrations. DO may be achieved the hypolipidemic effects by direct activation of hepatic Pparα expression and its responsive genes regulating hepatic fatty acid uptake and β-oxidation, while downregulating the hepatic fatty acid synthesis Our results demonstrate that high-fiber rice, DO, might be a potential supplement for the amelioration of insulin resistance and hyperlipidemia.
- Why choosing the heat treatment method for rice preparation?
- → There are several processes that increase RS in food technology. Representatively, it is retrogradation that occurs after heat processing (our manuscript also presents previous studies related to this in lines 78–80).
